

# Design and construction of an automated and programmable resistivity meter for shallow subsurface investigation

Antenor Oliveira Cruz Júnior[1], Cosme Ferreira Pontes-Neto[1], André Wiermann[1]

[1]National Observatory, Rio de Janeiro, Brazil

*Correspondence to*: Antenor O C Júnior (antenorjunior@on.br)

**Abstract.** Geoelectrical resistivity is an excellent method to investigate the structural composition of shallow subsurfaces. However, existing commercial equipment is typically expensive and often requires proprietary accessories and software to provide full system functionality. The objective of this study was to develop a multichannel, modular, automated, and programmable geo-resistivity meter capable of customization and programming by the user. To this end, a conceptual

prototype was built based on free software and open hardware technologies as a low-cost alternative to commercial equipment, while maintaining the accuracy and quality of the data at the same level. The prototype was based on electrode multiplexing to make the switching process more efficient by reducing cabling complexity, while synchronous demodulation for signal detection was employed, providing strong rejection of spurious electrical noise, typical of urban areas where such equipment is frequently used. The results show the feasibility of this project and demonstrate an important academic

contribution to open-source instrumental research.

## 1 Introduction

The use of geophysical methods for geotechnical investigation has become increasingly necessary to map large areas in the subsurface, bringing a significant gain in time and cost compared to traditional direct investigations, such as drilling and excavations (Santos et al., 2021; Augusto Filho et al., 2016; Silva, 2016). This indirect investigation involves performing

measurements on or near the surface, which are influenced by the internal distribution of the physical properties of the minerals and rocks that constitute it (Kearey et al., 2009). Several authors (Parkhomenko, 2012; Keller, 1966; Zonge, 1972) have studied this theory through experiments relating the mineralogical composition, texture, and arrangement of rocks, imputed to their electrical properties, specifically their resistivity, magnetic permeability, and dielectric constant. From there, a geological section is defined as a medium whose existing materials present different properties under the stimulus of

electric charges (Braga, 2016). In this sense, geoelectric investigation has become capable of producing measurements resulting in images of lithological sections, in addition to their geometric parameters, such as thickness, depth, and direction in different soil conditions and with different electrode arrangements, being influenced only by the stratigraphic distribution of the soil or by structures immersed in the subsurface (Kearey et al., 2009). Research efforts over the last few decades have led to the development of resistivity meters known commercially as georesistivimeters. This equipment is fundamental for



geoelectric exploration campaigns; however, companies that manufacture such instruments have been concerned with the quality of their measurements as well as their robustness for field work (Clement et al., 2020; Pasquali et al., 2017; Toll, 1999), making this instrument sophisticated and expensive. In this regard, it is worth mentioning the work of Hassan (2014) and Clement (2020), who developed automated resistivimeters for low-cost investigations, to disseminate the method from a robust and flexible tool for small-scale experiments, using basic bench-top electronic equipment for testing and feasibility in

the conceptual development of the instrument. In this work, the preliminary results obtained with the application of electroresistivity are reported, with the construction of a prototype of a modular professional resistivimeter, automated and programmable, aiming to prove the viability in the form of a proof of concept. Several aspects of design and construction have been applied to minimize the cost of construction and optimize the operation of the instrument in geophysical surveys based on some deficit aspects of commercial equipment (Silva, 2016).


## 2 Numerical Method

The electroresistivity method is based on the fact that different materials, whether geological or not, present different electrical resistivity values $(\rho)$ intrinsic to their constitution and proportion. This electrical conduction phenomenon depends on the mineralogical composition of the rocks as well as other factors that influence the final resistivity value, such as

mineralogy, porosity permeability, pressure, temperature, and the nature of the fluid in the pore. In this context, knowledge of the physical laws related to the electrical properties of geological materials becomes indispensable, considering that their propagation occurs in a three-dimensional medium (Orellana et al., 1972). The literature indicates that the conduction of electric current in the subsurface can occur in three distinct ways: electronic in substances with free electrons, dielectric in poor conductors or insulators, and ions (ionic or electrolytic conduction), which is undoubtedly the conduction process of the

greatest importance and interest in prospecting by electroresistivity (Menezes, 2013; Luiz et al., 1995; Feitosa et al., 2008). The working principle of the DC resistivity method and operation is mostly performed using a quadrupole device (Figure 1), where electric current is injected into the soil through insulated cables connected to an electric generator equipped with an ammeter to electrodes A and B, and likewise, cables are attached to a voltmeter or potentiometer capable of measuring the potential difference between electrodes M and N simultaneously (Dong et al., 2021; Menezes, 2013).

If we perform the relevant measurements and bring the data and, in practice, the current flow in a subsurface geological structure does not behave as in a homogeneous medium, this will result in a dummy resistivity $\rho_a$, which commonly will not be equal to $\rho_1$ or $\rho_2$ but will depend on $\rho_1$, $\rho_2$, $\rho_3$ and $\rho_4$ and the distances separating the electrodes. This resistivity cannot be the same as the average or weighted average of the four resistivities present, and may even be greater or less than all of them. This fictitious resistivity $\rho$ obtained by applying to the data obtained in a heterogeneous medium, the expression

corresponding to a homogeneous medium, is known as apparent resistivity; that is, this calculated value represents the apparent electrical resistivity of a homogeneous medium that is replaced by a heterogeneous medium, would cause the same



electrical reactions observed under the same geometric conditions of electrodes A, B, M, and N (Sato, 2002; Keller et al., 1966), to detect the presence of anomalous conductivity (Equation 1).

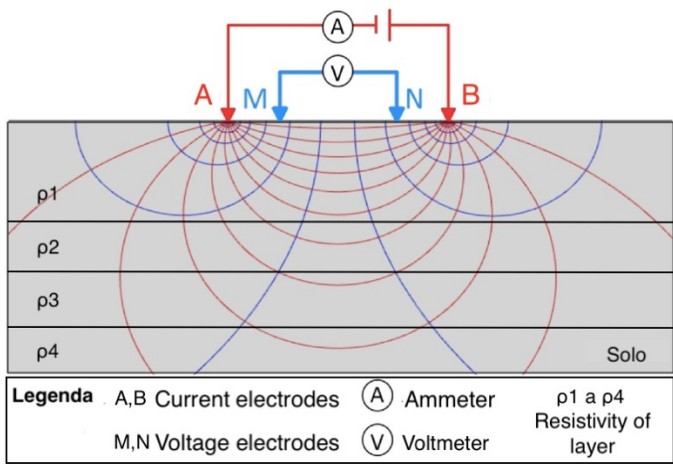

**Figure 1: Quadrupole electrode configuration through direct ground contact made by metallic or porous electrodes.**

$$\rho_a = k\left(\frac{\Delta V}{I}\right) (1)$$

where k is the geometric factor that depends on the quadrupole arrangement described in Equation 2, as follows:

$$k = 2\pi\left(\frac{1}{AM} - \frac{1}{BM} - \frac{1}{AN} + \frac{1}{BN}\right)^{-1} (2)$$

## 3 Methods

The basic concept of this electronic project is to develop equipment capable of performing all the necessary functions for the analysis and interpretation of a physical parameter using the electroresistivity method. The instrument structure block diagram (Figure 2), composed of a platform with modular hardware (composed of three modules), is aimed at the implementation of independent technologies, facilitating the acquisition, maintenance, and customization of the instrument in order to maximize its performance while staying within the constraints of cost, power consumption, and component availability. Descriptions of the modules and their functionality are as follows:




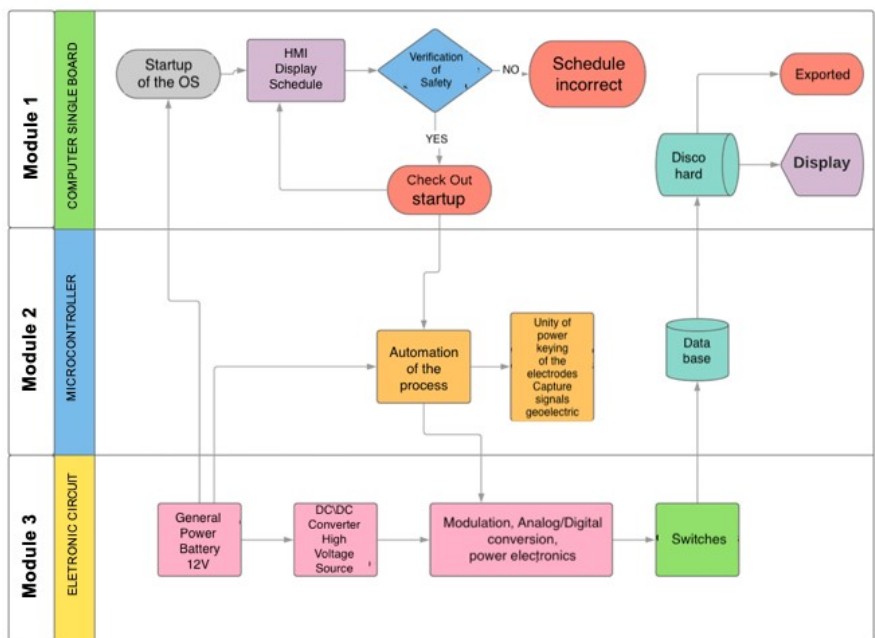

**Figure 2: Architecture and organization of the internal structure of the electrical resistivity instrument.**

## 3.1 Computational Unit

This unit is responsible for all operationalizations and configurations of the prototype. It consists of a single-board computer (SBC), among which we highlight the system initialization, human-machine interface (HMI), security drive, visualization, I/O mechanisms, integration with other sensors, serial communication, sensors, serial communication, and pre-processing of geoelectric signals.   To perform many activities, the SBC has an operating system based on the Linux Kernel, uses a BCM2711 ARM cortex A72 quad-core 1.5 GHz microprocessor, and 5V DC (3A) power supply. One of the most powerful features of the SBC is its input/output GPIO bus, which allows communication between electronic devices connected to the board, such as LCDs, various sensors, automation, and peripherals.   As shown in Figure 3, the prototype has a 7" TFT LCD with an 800 × 480 pixel resolution. Another promising aspect of this prototype is its human-machine interface (HMI) or user interface (Figure 3-a), which directs the input and output of information in the system. This HMI enables the user to configure the system and visualize the data without the need for an auxiliary computer (Figure 3-b). This powerful SBC allows two important additional features. First, it allows the programming of new electrode arrangements in a programmable and automated manner, allowing the equipment to be used in the research of new methodologies in geoelectrical surveys. Conventional arrangements with four electrodes are very practical for the interpretation of results, but the possibility of making many combinations between all electrodes in multichannel equipment allows the acquisition of a very large amount of linearly independent data, allowing the study of new methods of data inversion and interpretation. In addition, a preliminary interpretation of the data with graphical representation is possible, allowing the verification of the sanity of the





raw data in the field and eliminating or at least minimizing the capture of compromised data, which often occurs because of failures in the electrode connections to the ground.

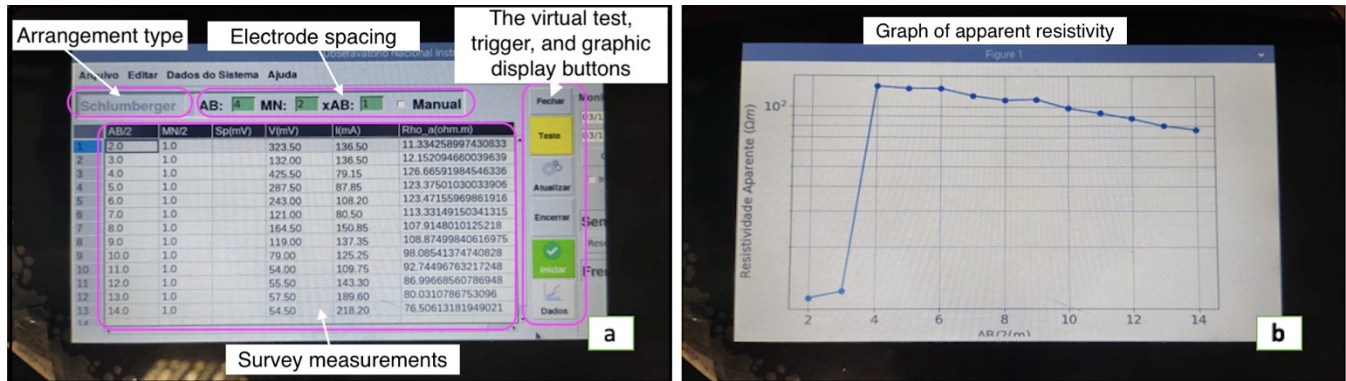

**Figure 3: Computer unit running man-machine interface for georesistivity meter configuration demonstrating the capture of geoelectric data (a). Graphical implementation demonstration based on data obtained in a field survey, relating apparent resistivity $\rho_a$ to AB/2 injection electrode openings (b).**

## 3.2 Process Automation Unit

This was responsible for automating the platform. This unit has an integrated circuit that gathers a processor core, volatile and non-volatile memories, programmable serial UART communication, an I²C SPI interface, a 6-channel A/D converter, and 10-bit resolution (Figure 4). This microcontroller has firmware designed to control various functions, such as control of the power unit and switching of the electrodes, depending on the arrangement used, capture, and temporary storage of the geoelectric signals. The option of using this component is its versatility and cost, and it is easily replaceable in the case of damage.



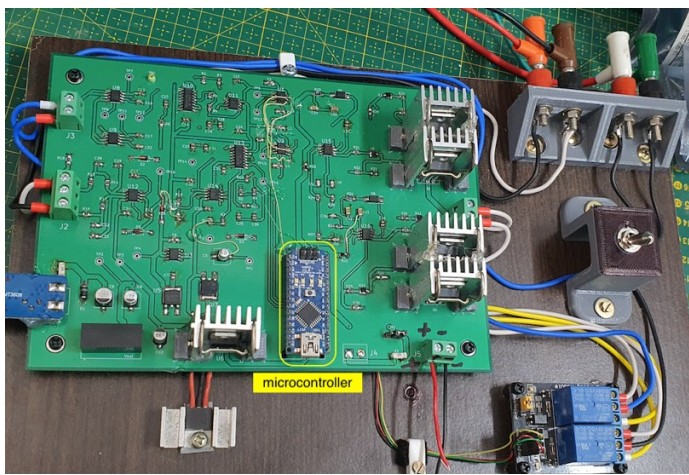

**Figure 4: Microcontroller coupling via female pin bar to the electronic circuitry to facilitate eventual board replacement.**

## 3.3 Electronic Circuitry

The electronic circuit diagram (Figure 5), shows all the main functions of the prototype. The voltage and current signals received from the front-end circuit (electrodes, differential buffers, and current sensors, Figure 5-c) were fed into a synchronous demodulation circuit that rectifies and delivers DC levels to a pair of analog-to-digital converters (Figure 5-b).

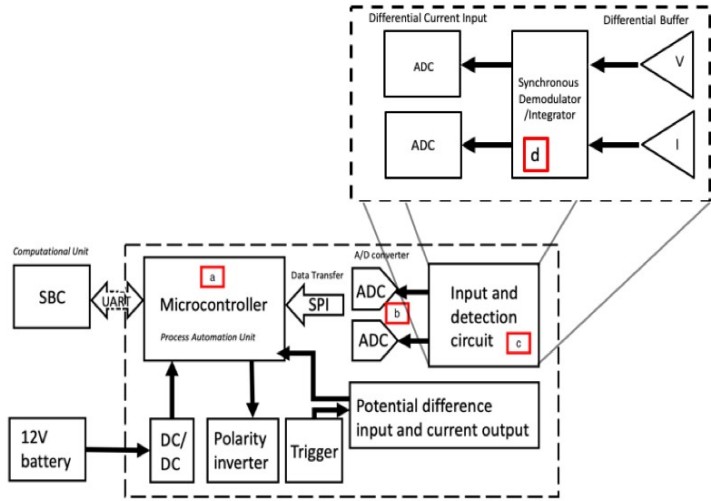

**Figure 5: Schematic of the main circuit components developed for the prototype.**



At the differential voltage input, the geoelectric voltage from the receiving electrodes is buffered and applied to a radio-frequency filter to mitigate any high-frequency interference. To accurately measure small differential voltages, an INA149 differential amplifier was placed in this circuit, which can tolerate common-mode signals in the range of ± 275 V. Inputs are protected from momentary common mode or differential overloads of up to 500 V. The measurement of the current injected into the ground was initially made using an ACS712 transducer chip, but due to its low sensitivity that in bench test reproduced a strong noise at very low currents, we opted for a pair of LTS 9-NP closed-loop multirange transducers that use Hall effect technology for the electronic measurement of currents (DC, AC, pulsed, mixed) in a wide frequency range (various waveforms) and with full galvanic isolation between the primary and secondary circuits. This transducer uses a unipolar voltage source from 0 to 5 V. Two transducers were placed in a differential configuration, doubling the overall sensitivity and creating a pseudo-differential signal with a 2.5 V common level. The analog-to-digital converters (Figure 5-b) have a 22-bit resolution in the Delta-Sigma architecture and a 15 s sampling rate, which communicate via a serial SPI interface to the microcontroller (Figure 5-a). As the geoelectric method is a process of successive electrical current injections followed by voltage registration, these signals are acquired and temporarily stored in the memory of the microcontroller for posterior transmission to the main CPU (SBC). The platform uses a 12-volt vehicle battery as a source, which allows it to supply all the voltages necessary for the operation of the system. From this battery, a DC/DC converter circuit under the microcontroller command generates several different voltages and frequencies to drive soil current excitation. The use of AC excitation and synchronous demodulation is essential to eliminate electrolytic bias and spurious noise that commonly affect geoelectric signals. An electromechanical switching array, also under the microcontroller command, enables the operation of any category of electrical probing technique in a linear quadripole system.

**4 Test**

The field test results were generally consistent with the objectives of this study. The figures are generally in line with the objectives of the work to develop a prototype concept with a differentiated characteristic capable of providing accurate measurements with low noise suitable for many field techniques commonly used in geophysical surveys. To test the accuracy of the measurements, a basic bench circuit was used to compare the measured current and potential of the electrodes. In these field tests, vertical electric sounding (SEV) was used with the Schlumberger arrangement and twenty-four electrodes arranged in a line, working at a voltage of 100 volts. This field test with a traditional bench circuit made it possible to obtain a calibration curve (Figure 7) and current and voltage measurements that could be compared with those obtained with the prototype in tests at the same site (Figure 6). The site chosen for testing was located within the National Observatory premises in the state of Rio de Janeiro. Studies carried out at this site through boreholes indicate the presence of a shallow subsurface landform followed by a sandy-silt horizon, fine to medium, interpreted as residual soil.





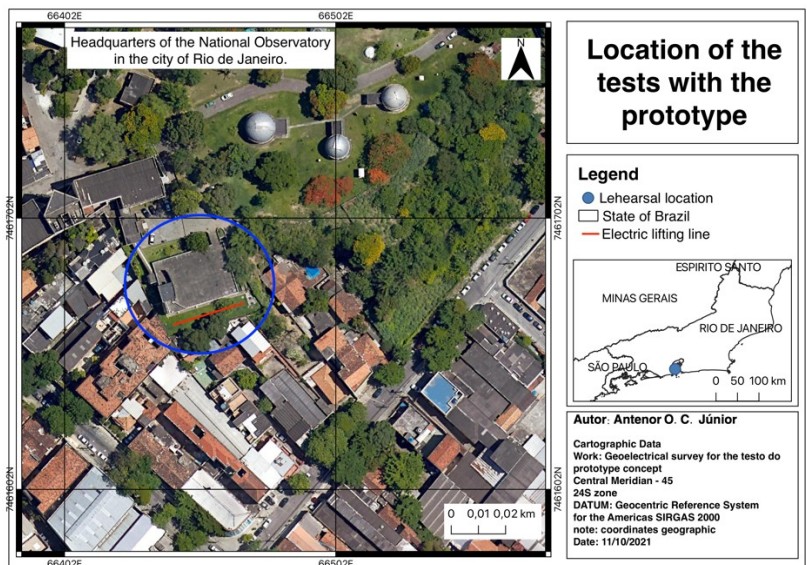

**Figure 6: Survey site for testing the prototype ( © Google Earth, 2021.)**

## 5 Results and discussions

The measurements obtained in the field were, for the most part, consistent with the objectives proposed in the work, qualitatively and quantitatively evidencing the good functioning of the developed low-cost prototype for obtaining in situ resistivity parameters at small depths. From this perspective, the tests involving the voltage and current presented good performance and accuracy based on the observed results, showing a linear correlation obtained by the ADCs of the prototype compared to the measurements of electronic equipment to measure these electrical quantities with high accuracy.

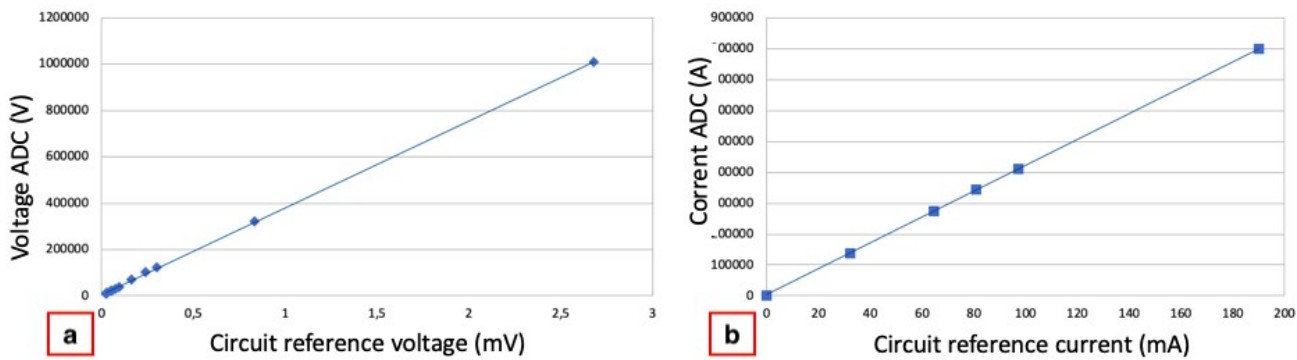

**Figure 7: Linear correlation graph showing the field test results comparing the voltage (a) and current (b) measured by the prototype against those obtained by a basic bench-top circuit for current and potential difference measurement through a qualitative survey conducted in the laboratory.**


The regression analysis that correlated the current found y = 4192.3x + 3375.3 and $R^2$ = 0.9998, where y is the measurement obtained by the prototype and x is the measurement obtained with an ammeter (Figure 7-a), whereas the voltage measurements had an equal regression y = 375353x + 4383 and $R^2$ = 0.9998 (Figure 7-b). Differential mode measurements

and synchronous demodulation were the appropriate choices, ensuring reasonably linear and reliable measurements. By analyzing these same measurements, we observed that the average percentage variation between the two pieces of equipment was 9.4% (Figure 8). Generally, the resistivity distributions along the section are comparable, even with some visible variations passive to be attributed to environmental effects due to the rainy season of the period to which this test was conducted, even though voltages of several millivolts, coming from 60 Hz in the surroundings of the test area, were not

detected in the measurements made by the prototype, only affecting estimations of the apparent resistivity of the soil.

The equipment proved to be easy to handle, given that all the functions are well described in its control interface, and amenable to improvements owing to its open code that allows customization of the entire operational process of the

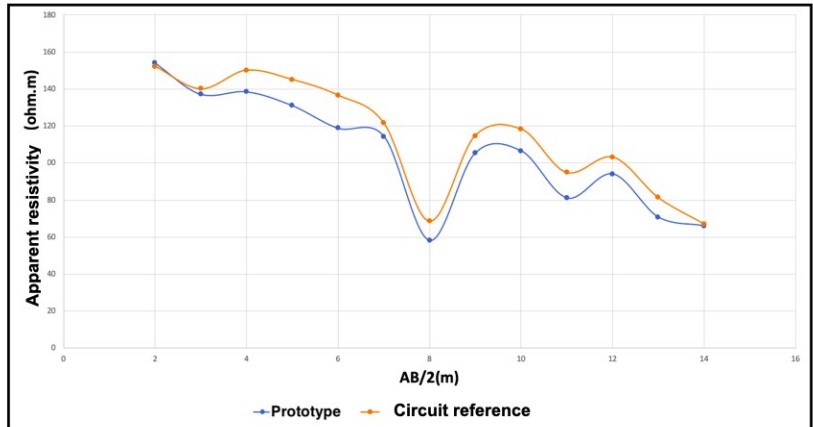

equipment by the user. Another important aspect is in its current configuration that, for this prototype, allows it to deliver higher stresses, thus enabling deeper soundings.

**Figure 8: Apparent resistivity test results obtained for a vertical electrical survey on a Schlumberger arrangement. Comparison of prototype data with resistivity measurement by bench instruments.**

Table 1. Main features of this project, showing relevant aspects derived from its original concepts.

| Appearance | Description |
| --- | --- |
| Modular Hardware | Creates the possibility of hardware upgradability and implementation of easy-to-acquire technologies. |
| Programmable | Open Source software that allows customization of the system. |
| Automation | Multiplexing of electrodes in an efficient manner with switching and reduced amount of relays distributed along the cable. |
| AC electrical signal | Using an electronic active circuit made it possible to circumvent the problems of |





| | spontaneous potential and ground polarization. |
|---|---|
| Programmable Frequency | Enables the use of different frequency values that can reach hundreds of Hertz. |
| Differential amplifiers | Avoid electrostatic discharge or undue induction to ground. |
| Computer unit | It allowed the implementation of a graphical interface, as well as the possibility of running three-dimensional resistivity modeling and inversion algorithms for data processing comparable to those of commercial instruments, making this prototype unique regardless of the need for an auxiliary computer. |
| Synchronous demodulation | Used so that the measured output voltage is a reflection of the excitation current, as it is concerning frequency and phase, thus eliminating spurious noise, noise not correlated to the excitation source. |

## 6 Conclusions

In this work, we constructed a functional prototype, with the objective of achieving the performance of a professional instrument, as an open-source hardware and software solution, based on easily accessible components and modules, aiming to provide equipment capable of being used in teaching and geophysical research.

The field survey experience confirmed the feasibility of using multielectrode systems and automated acquisition owing to the implementation of customized code through the programming of new electrode arrangements for this type of survey. The use of an AC electrical signal assisted by a synchronous demodulation circuit allowed us to circumvent the problems of spontaneous potential, earth polarization, and electrical induction from the power grid. Some initiatives exist for the production of prototypes of geoelectric instruments, but they are usually limited to didactic demonstration or proof of concept, without significant instrumental performance. Our modular hardware prototype brought a robust computational capacity to the proposed equipment, including a graphical interface that brings the possibility of implementing new functions and sensors, in addition to executing resistivity and inversion modeling algorithms for the treatment of data comparable to commercial instruments, as shown in Figure 9. In the near future, all circuits and software developed in this study will be made available to the scientific community on a website, allowing their free use and eventual contributions to the project.

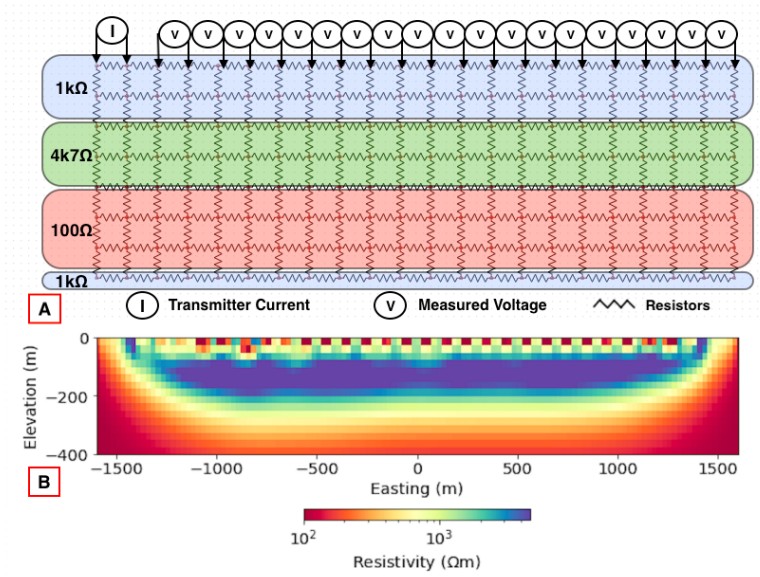

**Figure 9: Simulated inversion for a dipole–dipole arrangement in the resistive layer model format (a) shows that the implementation of the measurement treatment can be performed by the equipment in loco with good performance, even though no configurations or adjustments were made to improve the results used for testing the programming and automation processes. In this validation test, we considered only three horizontal layers of the mesh of resistors with values of 1000$ \Omega $m, 4700$ \Omega $m, and 100$ \Omega $m. It is possible to identify clear features, such as the separation of layers (b), vertical discontinuity,**
**and order of magnitude of the resistivity, which are compatible with the values used in the test.**

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
