# Peer review of "Design and construction of an automated and programmable resistivity meter for shallow subsurface investigation"

_Geoscientific Instrumentation, Methods and Data Systems, 2022_

## Author Response (AR1)

Dear Associate Editor

These are the point-by-point answers to the comments

Thanks in advance for the opportunity and input from the reviewers, I appreciated the constructive criticism and addressed each of your concerns as outlined in the discussion contributions. I enclose a pdf version of the manuscript GI-2022-2, with the corrections made in colored highlights, to facilitate the conference. The corrections made and arguments are shown below, item by item, as suggested.

Text revisions (by line number)

32-35: Maybe this should be split into two sentences, it's hard to read with so many commas and separate sentences.

Answer = answered. An unfolding of the paragraph was made for better understanding of the text in the introduction. This adjustment is highlighted on a yellow background (line 33).

43-45: "...mineral composition... as well as other factors... such as mineralology..." may be redundant.

Answer = answered. Unfortunately, this point was not clear in our original article. We would like to apologize for the misunderstanding and we are now going to revise the article to explain this better. This setting is marked on a yellow background (lines 41 to 50).

55-56: The term "bring the data" is confusing here.

Answer = answered. Modified as per guidance. This setting is marked on a yellow background (lines 58 to 62).

59-60: "..applying the expression corresponding to a homogeneous medium to the data obtained..."

Answer = answered. We modified a, as directed. This setting is highlighted in yellow background (line 55).

61: "by a heterogeneous medium." This sentence is too long and doesn't benefit from the rest of the exposition after this point.

Answer = answered. We modified according to guidance. This setting is marked on a yellow background (lines 57 to 61).

107-108: "...a 6-channel A/D converter and 10-bit resolution."

Answer = answered. We would like to apologize for the misunderstanding and now we are going to review this information about the analog-to-digital conversion. In the project we used the MCP3550 (ADC) delta-sigma of 22 bits of resolution. This adjustment is highlighted in yellow background (line 108 to 109).

110: This is a sentence fragment.

Answer = answered. We believe that the discussion has become clearer, and consequently the conclusion more consistent. This setting is marked on a yellow background (lines 106 to 108).

173-174: Higher stresses meaning higher excitation voltages

Answer = answered. Unfortunately, this point was not clear in our original article. This paragraph deals with an important aspect conceived with the idealization of the project to allow the use of higher voltages that were used in the controlled tests, allowing deeper soundings. After the reformulation made with the evaluator's contributions, we believe that the discussion has become clearer. This setting is highlighted on a yellow background (lines 173 to 174).

Formula 1: Must match Formula 2 formatting (center)

Answer = answered. Formula 1 was centered in the text, as directed. This setting is highlighted in yellow (line 66).

A link to the open source library (github, etc) containing software/hardware descriptions is highly recommended.

Answer = Regarding the disclosure of a link to the open source library containing software/hardware descriptions, we thank the reviewer for this valuable suggestion and it is in our interest that this equipment be made available to others surveyed, a subject that is already a consensus among the creators of the project. As soon as we have new advances, we will be able to start fully publicizing the project.

Section 1 (Introduction) (lines 24-29) describes the use of resistivity meters in geological exploration. Here, I suggest adding literature references on recent advances in testing time-lapse resistivity tomography to monitor shallow subsurface saturation changes.

Answer = At first, we chose to focus on the development of electronics and measurement accuracy in simple geophysical surveys, to test and evaluate the conceptual idea to prove its feasibility. Following your suggestion, we added some references on time-lapse resistivity. We agree that it is a relevant topic and we intend to address this issue in more depth in the field test that the equipment will soon undergo.

Most importantly, at the end of the introduction, the manuscript would greatly benefit from a concise description of the concrete goals in terms of cost, power consumption and component availability that they aim to achieve and a discussion of how these instruments can complement existing instruments in terms of performance.

Answer = It is in our interest that this equipment be made available to other researches, a subject that is already a consensus among the creators of the project, however, the availability of information about the process inherent to the logistics of construction, as well as cost, consumption and availability of components of the equipment have not yet been raised, since, the main object was to validate the feasibility of a concept.

Section 3.1 "Computational unit" states that the flexibility of the instrument would allow testing new methods of data inversion (lines 93 - 95). I strongly encourage supporting this with literature references and discussing advantages and disadvantages of conventional instruments. This comment is related to my comment on the introduction and could possibly be addressed together.

Answer = The project is intended to contribute to the development of open instrumentation and programming Geophysics, which will enable the implementation of new investigation techniques or arrangements for the geoelectric method. Unlike commercial equipment, this platform will allow a greater number of configurations in its software and the use of algorithms (developed by the researcher himself or by other collaborators), in the treatment of the data obtained. It is worth noting that there are already works that address the same theme that were cited in the text of the article, for low-cost, robust and flexible investigations, for small-scale experiments using state-of-the-art electronic equipment.

I encourage a clearer distinction between methods, results, and discussion. I read section 4 ("Testing") as an extension of methods. The sentences in line 142-144, however, begin with an interpretation and are redundant with what follows in section 5.

I understand that the focus of this paper is on hardware and software engineering. However, the interpretation of the field test seems a bit short. The subsurface characteristics of the study site at the National Observatory are mentioned in line

149. I suggest that additional information from previous resistivity surveys at this site be provided and discussed along with the results obtained with this prototype.

Answer = The entire section has been redrawn. In fact, confusion occurred when drafting the text. This adjustment is marked in yellow background (lines 143 to 149).

Later, in line 167, the environmental effects (rainy season) are discussed, which should also be related to Figure 9, which is not referred to here. I recommend outlining how a second test during a different soil saturation situation could be performed. Also, the sentence in line 167 is long and should be rephrased in a more understandable way.

Answer = New tests are scheduled plus that will not be able to happen in time to be able to be incorporated into the original article.

Other technical corrections:

Section 2 (Numerical methods): The sentence (line 55) is very difficult to understand. The sentence should be clarified and possibly split in two. The sentence (line 59 onwards) should also be simplified and split, as it is difficult to understand.

Answer = answered. Modified, as directed. This adjustment is marked on a yellow background (lines 58 to 65).

All the analysis and plotting functions that the human-computer interface provides, of which some, are shown in Figure 3, mentioned in this manuscript? They can also be shown in a diagram.

Answer = A diagram for the HMI is now included. The online repository for this project to be published soon will also contain all source codes, diagrams and function descriptions.

The order of the figures should be improved to match the text. Figure 9 showing the arrangements should be shown together with Figure 6.

Answer = complied with. Figure 9 demonstrates the prototype's ability to run modeling algorithms in the Python language that made it possible to produce the data treatment. This adjustment is marked on a yellow background (lines 189 to 192).

Figure 6 would benefit from a more detailed legend description, providing more information on how to read the schema.

Answer = We have modified as directed. This adjustment is marked on a yellow background (lines 151 to 152).

Figure 7 requires adjustments regarding the readability of the y-axis, as well as a spell check.

Answer = complied with. We have modified as directed.

Figure 9: Some units are not displayed correctly in the legend. Should it really be "4k7â"¦"?

Answer = understood. We would like to apologize for the misunderstanding and will now review this unit of measure information. This setting is marked on a yellow background (line 198).

Figures/Tables

Fig. 1: Subtitle typo

Answer = answered. We modified according to guidance. Figure 1.

Fig. 2: It's unclear what "Hard Disk" refers to

Answer = answered. We modified according to guidance. Figure 2.

Fig. 5: The callout makes it look like the demod/integrator (d) synch feeds two stages of ADCs. Is this accurate?

Answer = Yes. The demodulator and associated integrator circuit have two channels operating in tandem.

Fig. 7a: It is difficult to say whether there are significant variations in the lower values due to the scale of the graph. It may be helpful to present this on a logarithmic scale, although recognizing that what is being presented is linear correlation. Perhaps just omitting the upper end of the plot is better. The use of engineering notation is encouraged.

Answer = answered. We modified according to guidance. Figure 7a.

Fig. 7b: The y-axis values are crowded by the y-axis label and are unreadable. The use of engineering notation is encouraged.

Answer = answered. We modified according to guidance. Figure 7b.

Fig. 9: This plot is not well motivated in the text or in the caption. I would like to see at least a paragraph explaining the figure as well as the software that generated this figure so it remains in manuscript.

Answer = answered. The prototype has the ability to execute modeling algorithms in Python language that made it possible to produce the treatment of synthetic test data, as shown in Figure 9. This adjustment is highlighted in yellow background (lines 189 to 192).